# Pretreatment tumor-related leukocytosis misleads positron emission tomography-computed tomography during lymph node staging in gynecological malignancies

Seiji Mabuchi[1,2,5 ✉], Naoko Komura[1,5], Tomoyuki Sasano[3], Kotaro Shimura[1], Eriko Yokoi[1], Katsumi Kozasa[1], Hiromasa Kuroda [1], Ryoko Takahashi[1], Mahiru Kawano[1], Yuri Matsumoto[1], Hiroki Kato [4], Jun Hatazawa[4] & Tadashi Kimura[1]

The accuracy of fluorine-18-fluorodeoxyglucose positron emission tomography-computed tomography (18F-FDG-PET/CT) can be influenced by the increased glycolytic activity of inflammatory lesions. Here, using clinical data obtained from gynecological cancer patients, tumor samples and animal models, we investigate the impact of pretreatment tumor-related leukocytosis (TRL) on the diagnostic performance of 18F-FDG-PET/CT in detecting pelvic and paraaortic lymph node metastasis. We demonstrate that pretreatment TRL misleads 18F-FDG-PET/CT during lymph node staging in gynecological malignancies. In the mechanistic investigations, we show that the false-positive 18F-FDG-PET/CT result for detecting nodal metastasis can be reproduced in animal models of TRL-positive cancer bearing G-CSF expressing cervical cancer cells. We also show that increased 18F-FDG uptake in non-metastatic nodes can be explained by the MDSC-mediated premetastatic niche formation in which proinflammatory factors, such as S100A8 or S100A9, are abundantly expressed. Together, our results suggest that the MDSC-mediated premetastatic niche created in the lymph node of TRL-positive patients misleads 18F-FDG-PET/CT for detecting nodal metastasis.

[1] Department of Obstetrics and Gynecology, Osaka University Graduate School of Medicine, Suita, Osaka, Japan. [2] Department of Obstetrics and Gynecology, Nara Medical University, Kashihara, Nara, Japan. [3] Department of Gynecologic Oncology and Reproductive Medicine, University of Texas MD Anderson Cancer Center, Houston, TX, USA. [4] Department of Nuclear Medicine and Tracer Kinetics, Osaka University Graduate School of Medicine, Suita, Osaka, Japan. [5] These authors contributed equally: Seiji Mabuchi, Naoko Komura. ✉email: smabuchi@gyne.med.osaka-u.ac.jp

Accurate lymph node staging is highly important for determining the optimal treatment strategy and for predicting the treatment outcomes in cancer patients. False-positive detection of distant nodes can result in unnecessary surgical resection or radiotherapy to the suspected nodes, which can cause increased toxicities. Moreover, false-positive detection of metastatic nodes may promote unnecessary administration of systemic chemotherapy and potentially result in the denial of a chance for surgical or radiological cure.

Computed tomography (CT) and magnetic resonance imaging (MRI) have been widely used to assess lymph node metastases in patients with gynecological cancers. Both techniques base the identification of metastatic nodes on node size measurements; a short-axis diameter greater than 10 mm is the most accepted criterion. However, the size of lymph nodes does not always correlate with their tumor involvement. Consequently, these morphological imaging techniques have low sensitivity and cannot replace lymphadenectomy; the sensitivity for detecting lymph node metastasis in gynecological cancers is 28–77% with a specificity of 78–100%[1]. Thus, a non-invasive modality that enables more accurate lymph node staging is needed.

Positron emission tomography (PET) with fluorine-18-fluorodeoxyglucose as a tracer (18F-FDG-PET) is a relatively new functional diagnostic technique based on the rationale that rapidly dividing malignant cells have an increased glucose metabolism, allowing for the detection of areas with cancer cells. To overcome the inherent disadvantages of 18F-FDG-PET (i.e., poor anatomical information), integrated 18F-FDG-PET/CT was developed. On the basis of reported diagnostic advantages over conventional imaging modalities[2–4], 18F-FDG-PET/CT has been widely used for staging, determining the extent of surgical resection, or planning the radiation fields in the management of gynecological cancers. However, the accuracy of 18F-FDG-PET/CT has been challenged by the increased glycolytic activity of inflammatory lesions because inflammatory cells, such as neutrophils, macrophages, and lymphocytes, exhibit increased 18F-FDG uptake[5–7]. In line with this, previous studies have suggested that virus infections (i.e., HIV, EBV, HCV), toxoplasmosis, tuberculosis, and immunizations (i.e., influenza vaccination) can cause significant 18F-FDG-uptake in lymph nodes mimicking malignant metastatic malignancy[8–13].

Systemic inflammatory reactions, including thrombocytosis, leukocytosis, neutrophilia or lymphocytopenia, have recently gained attention as indicators of poor prognosis or predictors of poorer treatment response in cancer patients[14,15]. Tumor-related leukocytosis (TRL) is a paraneoplastic syndrome that is occasionally encountered in patients with malignant tumors[16]. According to recent reports, TRL is observed in 5–10% of patients with gynecologic cancer and is associated with a poor prognosis[17–19]. We recently demonstrated that the "premetastatic niche", an immunosuppressive and proinflammatory environment, is generated in premetastatic organs by myeloid-derived suppressor cells (MDSC) to facilitate tumor cell metastasis in uterine cervical and endometrial cancer patients that exhibit TRL[20]. Considering the proinflammatory nature of the premetastatic niche, theoretically, MDSC-mediated premetastatic niches in TRL-positive cancer patients may manifest as hypermetabolic lesions on 18F-FDG-PET/CT, which can cause false-positive 18F-FDG-PET/CT results. However, the association between the presence of pretreatment TRL and the diagnostic performance of 18F-FDG-PET/CT during lymph node staging has yet to be investigated.

In the current study, using clinical data obtained from gynecological cancer patients, we first evaluated the impact of pretreatment TRL on the diagnostic performance of 18F-FDG-PET/CT in detecting pelvic and paraaortic lymph node metastasis. Then, using tumor samples obtained from gynecological cancer patients as well as animal models of gynecological cancers, we performed mechanistic investigations focusing on the MDSC-mediated premetastatic niche and 18F-FDG-uptake. Our study demonstrated that TRL can mislead 18F-FDG-PET/CT during lymph node staging in gynecological cancer patients.

## Results

**Patients.** A total of 551 gynecological cancer patients (426 in the primary cohort and 125 in the validation cohort) who underwent pretreatment 18F-FDG-PET/CT scans were included in the current study. The clinicopathological characteristics of these patients are summarized in Table 1 and Supplementary Table 2.

**Analyses in the primary cohort.** In the primary cohort, 127 patients had cervical, 203 had endometrial, and 96 had ovarian cancer. Pretreatment TRL (WBC ≥ 9000/μL) was observed in 17 (13.4%), 17 (8.4%), and 12 (12.5%) of the cervical, endometrial, and ovarian cancer patients, respectively (Table 1).

**Table 1 Clinicopathological characteristics of the patients in primary cohort.**

|  |  | All patients (n = 426) | Cervical cancer (n = 127) | Endometrial cancer (n = 203) | Ovarian cancer (n = 96) |
|---|---|---|---|---|---|
|  |  | n (%) | n (%) | n (%) | n (%) |
| Age (years) | <50 | 145 (34.0) | 73 (57.4) | 39 (19.2) | 33 (34.3) |
|  | ≥51 | 281 (66.0) | 54 (42.5) | 164 (80.8) | 63 (65.6) |
| FIGO stage | I | 263 (61.7) | 95 (74.8) | 114 (56.2) | 54 (56.3) |
|  | II | 58 (13.6) | 32 (25.2) | 19 (9.4) | 7 (7.3) |
|  | III | 97 (22.8) | 0 | 67 (33.0) | 30 (31.3) |
|  | IV | 8 (1.9) | 0 | 3 (1.5) | 5 (5.2) |
| Histology | SCC | – | 63 (49.6) | – | – |
|  | EAC | – | – | 167 (82.3) | – |
|  | SAC | – | – | – | 24 (25.0) |
|  | Others | – | 64 (50.4) | 36 (17.7) | 72 (75.0) |
| Peritoneal cytology[a] | Negative | 330 (77.5) | 127 (100) | 155 (76.4) | 48 (50.0) |
|  | Positive | 91 (21.4) | 0 (0) | 44 (21.7) | 47 (49.0) |
| WBC count (/μL) | ≥9000 | 46 (10.8) | 17 (13.4) | 17 (8.4) | 12 (12.5) |
|  | <9000 | 380 (89.2) | 110 (86.6) | 186 (91.6) | 84 (87.5) |

FIGO International Federation of Gynecology and Obstetrics, SCC squamous cell carcinoma, EAC endometrioid adenocarcinoma, SAC serous adenocarcinoma, WBC white blood cell
[a]Peritoneal cytology was not indicated for 4 cases in endometrial cancer and 1 case in ovarian cancer.

**Table 2 The impact of pretreatment TRL on the diagnostic performance of 18F-FDG-PET/CT during LN staging in gynecological cancer patients (primary cohort).**

| | | Cervical cancer (n = 127) | | | Endometrial cancer (n = 203) | | | Ovarian caner (n = 96) | | |
|---|---|---|---|---|---|---|---|---|---|---|
| | | False-positive PET/CT results | | | False-positive PET/CT results | | | False-positive PET/CT results | | |
| | | Yes (n = 10) | No (n = 117) | P-value | Yes (n = 12) | No (n = 191) | P-value | Yes (n = 8) | No (n = 88) | P-value |
| WBC count (/µL) | <9000 | 4 (3.6) | 106 (96.4) | <0.0001 | 5 (2.7) | 181 (97.3) | <0.0001 | 3 (3.6) | 81 (96.4) | <0.0001 |
| | ≥9000 | 6 (35.3) | 11 (64.7) | | 7 (41.2) | 10 (58.8) | | 5 (41.7) | 7 (58.3) | |

Statistical significance was assessed using two-sided Fisher's exact test.
*FDG* fluorodeoxyglucose, *PET* positron emission tomography, *CT* computed tomography, *WBC* white blood cell.

The 18F-FDG-PET/CT and histopathological findings are summarized in Supplementary Table 1. Overall, 18F-FDG uptake was present in 90.6% (115 of 127), 92.6% (188 of 203), and 88.5% (85 of 96) of primary tumors in cervical, endometrial, and ovarian cancer patients, respectively. Among the 127 cervical cancer patients, 23 (18.1%) exhibited increased 18F-FDG uptake in the retroperitoneal lymph nodes, indicative of metastasis. Pathological examinations after initial surgery suggested that 18F-FDG-PET/CT correctly diagnosed metastatic lymph nodes in 13 (10.2%) patients, but gave a false negative for 19 (15.0%) patients. 18F-FDG-PET/CT was correctly negative for 85 (66.9%) patients, and falsely positive for 10 (7.9%) patients (Supplemental Table 1). The sensitivity, specificity, negative predictive value (NPV), and positive predicitive value (PPV) of 18F-FDG-PET/CT in lymph node staging were 40.6%, 89.5%, 81.7%, and 56.5%, respectively. Similar results were obtained for patients with endometrial or ovarian cancer, consistent with previous reports[2–4].

Using clinical data obtained from patients, we next investigated the impact of pretreatment TRL on the diagnostic performance of 18F-FDG-PET/CT in detecting pelvic and paraaortic lymph node metastasis in gynecological cancer patients. As shown in Table 2, pretreatment TRL was significantly associated with false-positive 18F-FDG-PET/CT results (false-positive results in cervical cancer for TRL patients versus non-TRL patients; 35.3% vs 3.6%, $P <$ 0.0001, endometrial cancer; 41.2% vs 2.7%, $P < 0.0001$, ovarian cancer; 41.7% vs 3.6%, $P < 0.0001$). By multivariate analysis (Table 3), pretreatment TRL was found to be an independent predictive factor for false-positive 18F-FDG-PET/CT results (cervical cancer; odds ratio: 15.39; 95% confidence interval (CI), 3.48–79.23; $P = 0.0004$, endometrial cancer; odds ratio: 31.03; 95% CI, 6.57–146.53; $P < 0.0001$, ovarian cancer; odds ratio: 31.72; 95% CI, 4.45–225.98; $P = 0.0006$). These results strongly suggest the existence of certain factors that can cause false-positive 18F-FDG-PET/CT results in the lymph nodes (LNs) of TRL-positive gynecological cancer patients.

**Mechanistic investigations**. We previously demonstrated that increased granulopoiesis induced by tumor-derived G-CSF is responsible for the development of TRL[21], and that the pre-metastatic niche created by G-CSF-induced MDSC is responsible for the highly metastatic nature of TRL-positive uterine cervical and endometrial cancer[20]. By carefully reviewing the patients for whom 18F-FDG-PET/CT produced false-positive results, we noted that the majority of false-positively detected lymph nodes were significantly enlarged (Fig. 1a–d). Although 18F-FDG-PET/CT demonstrated increased 18F-FDG uptake in para-aortic lymph nodes (PALNs) (Fig. 1a, b), pathology of the resected enlarged lymph nodes revealed no malignant cells (Fig. 1c, d). Thus, we considered that the pro-inflammatory environment is created in the lymph nodes of TRL-positive patients (Fig. 2).

To further investigate whether TRL affects the diagnostic performance of 18F-FDG-PET/CT during lymph node staging, we employed a rat model of TRL-positive and TRL-negative cervical cancer, in which cervical cancer cells stably transfected with G-CSF or control vector were subcutaneously inoculated (Fig. 3a). The expression of G-CSF in these cells was confirmed in vivo (Fig. 3b). ME180-GCSF-derived tumor-bearing rats had significantly higher white blood cell counts and granulocyte counts than ME180-Control-derived tumor-bearing rats (Fig. 3c). Importantly, ME180-GCSF-derived tumor-bearing rats had significant 18F-FDG-uptake in the PALNs (Fig. 3d). As shown in Fig. 3e and f, the corresponding PALNs obtained from the same rats were significantly enlarged; however, pathological examination demonstrated no tumor involvement in these PALNs. In contrast, we detected no PALN 18F-FDG uptake or PALN enlargement in ME180-Control-derived tumor-bearing rats (Fig. 3e, f).

To investigate whether the premetastatic niche is created in the PALN falsely detected by 18F-FDG-PET/CT, we first investigated the number of MDSC in the PALNs of rats bearing ME180-Control-derived tumors or ME180-GCSF-derived tumors. Based on previous reports[22,23], rat MDSC were defined as CD11b/c$^+$ HIS48$^+$ cells. As shown in Fig. 3g, significantly higher numbers of MDSC were observed in the peripheral blood and PALNs of ME180-GCSF-derived tumor-bearing rats than in those of ME180-Control-derived tumor-bearing rats, suggesting that tumor-derived G-CSF induce MDSC from bone marrow.

As MDSC depletion can be achieved by anti-Gr-1 antibody in mice, we next investigated the effect of tumor-derived G-CSF on PALN and MDSC in the presence or absence of anti-Gr-1 antibody. Consistent with the findings in humans and rats, the PALNs in the ME180-GCSF-derived tumor-bearing mice were significantly enlarged when compared with those from ME180-Control-derived tumor-bearing mice (Fig. 3h). When the ME180-Control derived tumor-bearing mice or ME180-GCSF derived tumor-bearing mice were treated with the anti-Gr1 antibody, no enlarged PALNs were observed (Fig. 3h). Moreover, the number of CD11b$^+$Gr1$^+$ cells in the peripheral blood or the PALN were significantly higher in ME180-GCSF-derived tumor-bearing mice than ME180-Control-derived tumor-bearing mice, and the number of CD11b$^+$Gr1$^+$ cells was significantly decreased by the treatment with anti-Gr1 antibody (Fig. 3i). Importantly, as shown, CD11b$^+$Gr1$^+$ cells obtained from the mice showed significant suppressive activity on CD8$^+$ T cells (Fig. 3j), indicating the suppressive activity of MDSC.

Previous studies demonstrated that MDSC induced by tumor-derived G-CSF express S100A8/A9[20,24], and that S100A8/A9 expression in the premetastatic tissue is indicative of the MDSC-mediated premetastatic niche[20,25]. Thus, we investigated whether tissue S100A8/A9 expression in the absence of cancer cells can induce 18F-FDG-uptake. For this purpose, we subcutaneously

**Table 3 Univariate/multivariate analyses for false-positive 18F-FDG-PET/CT results.**

| Variables | | Univariate analysis | | | Multivariate analysis | | |
|---|---|---|---|---|---|---|---|
| | | Odds ratio | 95%CI | *P*-value | Odds ratio | 95%CI | *P*-value |
| Cervical cancer | | | | | | | |
| Age (years) | <50 | 1 | | | 1 | | |
| | ≥51 | 1.39 | 0.38–5.06 | 0.6193 | 2.52 | 0.55–12.96 | 0.2324 |
| FIGO stage | IA-IIA | 1 | | | 1 | | |
| | IIB | 0.41 | 0.05–3.38 | 0.4067 | 0.20 | 0.01–1.79 | 0.1637 |
| Histology | SCC | 1 | | | 1 | | |
| | Others | 0.66 | 0.18–2.44 | 0.5291 | 0.62 | 0.12–2.87 | 0.5439 |
| Tumor size (mm) | <40 | 1 | | | 1 | | |
| | ≥40 | 2.22 | 0.54–8.36 | 0.2554 | 1.84 | 0.30–9.80 | 0.4875 |
| WBC count (/μL) | <9000 | 1 | | | 1 | | |
| | ≥9000 | 14.45 | 3.53–59.16 | 0.0002 | 15.39 | 3.48–79.23 | 0.0004 |
| Endometrial cancer | | | | | | | |
| Age (years) | <50 | 1 | | | 1 | | |
| | ≥51 | 0.45 | 0.13–1.57 | 0.2107 | 0.59 | 0.10–3.41 | 0.5563 |
| FIGO stage | IA-IIB | 1 | | | 1 | | |
| | IIIA-IVB | 2.84 | 0.87–9.32 | 0.0842 | 2.31 | 0.41–13.24 | 0.3451 |
| Histology | EAC | 1 | | | 1 | | |
| | Others | 2.48 | 0.71–8.75 | 0.1565 | 4.11 | 0.80–21.10 | 0.0904 |
| Peritoneal cytology[a] | Negative | 1 | | | 1 | | |
| | Positive | 1.34 | 0.34–5.30 | 0.6722 | 0.98 | 0.15–6.55 | 0.9831 |
| WBC count (/μL) | <9000 | 1 | | | 1 | | |
| | ≥9000 | 25.34 | 6.82–94.14 | <0.0001 | 31.03 | 6.57–146.53 | <0.0001 |
| Ovarian cancer | | | | | | | |
| Age (years) | <50 | 1 | | | 1 | | |
| | ≥51 | 4.00 | 0.47–33.99 | 0.2042 | 5.62 | 0.42–75.29 | 0.1919 |
| FIGO stage | IA-IIB | 1 | | | 1 | | |
| | IIIA-IVB | 1.84 | 0.43–7.86 | 0.4114 | 3.28 | 0.37–28.91 | 0.2839 |
| Histology | SAC | 1 | | | 1 | | |
| | Others | 1.00 | 0.19–5.32 | 1.0000 | 3.27 | 0.31–34.05 | 0.3209 |
| Peritoneal cytology[a] | Negative | 1 | | | 1 | | |
| | Positive | 1.79 | 0.40–7.94 | 0.4462 | 1.92 | 0.29–12.83 | 0.5010 |
| WBC count (/μL) | <9000 | 1 | | | 1 | | |
| | ≥9000 | 19.29 | 3.79–98.07 | 0.0004 | 31.72 | 4.45–225.98 | 0.0006 |

Statistical significance was assessed using multiple logistic regression analysis.
*FDG* fluorodeoxyglucose, *PET* positron emission tomography, *CT* computed tomography, *FIGO* International Federation of Gynecology and Obstetrics, *SCC* squamous cell carcinoma,
*EAC* endometrioid adenocarcinoma, *SAC* serous adenocarcinoma, *WBC* white blood cell.
[a]Peritoneal cytology was not indicated for 4 cases in endometrial cancer and 1 case in ovarian cancer.

injected recombinant S100A8/A9 into mice. As shown in Fig. 4a, the intensity of 18F-FDG uptake was significantly increased by the S100A8/A9 injection., indicating that tissue S100A8/A9 expression in the absence of cancer cells can induce significant 18F-FDG uptake. We then investigated the expression of S100A8/A9 in the PALNs falsely detected by 18F-FDG-PET/CT. As shown in Fig. 4b, significant immunoreactivity for S100A8/A9 was observed in the PALNs from ME180-GCSF-derived tumors-bearing rats, which was in clear contrast to the results obtained from the PALNs from ME180-Control-derived tumors-bearing rats. Moreover, in an in vitro experiment, MDSC isolated from the PALNs of ME180-GCSF cell-derived tumor-bearing mice expressed high levels of *S100a8* and *S100a9* mRNA (Fig. 4c). Collectively, these results suggest that MDSC express S100A8/A9 and create the premetastatic niche in ME180-GCSF-derived tumor-bearing rats, leading to the false-positive detection of metastatic lymph nodes by 18F-FDG-PET/CT.

To exclude the possibility that findings are specific to ME-180 cell line, we have conducted additional experiments using an endometrial cancer cell line. As shown (Supplementary Fig. 1a–h), the results obtained from experiments using Ishikawa cells were consistent with those obtained from ME-180 cervical cancer cells.

In order to confirm whether the results obtained in animal investigations are representative of the clinical status, we performed immunohistochemical analyses using lymph nodes of cervical cancer patients obtained at initial surgery. As shown in Fig. 5a, significantly higher numbers of tumor-infiltrating MDSC (CD33+ cells) were observed in the lymph nodes from TRL-positive cervical cancer patients than in those from TRL-negative cervical cancer patients. Moreover, significantly greater immunoreactivity for S100A8/A9 was observed in lymph nodes from TRL-positive cervical cancer patients than in those from TRL-negative cervical cancer patients (Fig. 5b and 5c). In addition, as shown in Supplementary Fig. 2, more than 75% of false-positive lymph nodes showed strong immunoreactivities for CD33, S100A8, and S100A9, which were consistent with the results obtained in animal investigations. Furthermore, as shown in Supplementary Fig. 3, among the patients with false-positive lymph node, TRL-positive patients showed significantly higher G-CSF expression in their primary cervical tumor than TRL-negative patients.

**Analyses in the validation cohort**. Finally, to validate the findings obtained from clinical and preclinical investigations, we investigated the impact of pretreatment leukocytosis on the diagnostic performance of 18F-FDG-PET/CT in the lymph node staging of newly-diagnosed gynecological cancer patients. In the

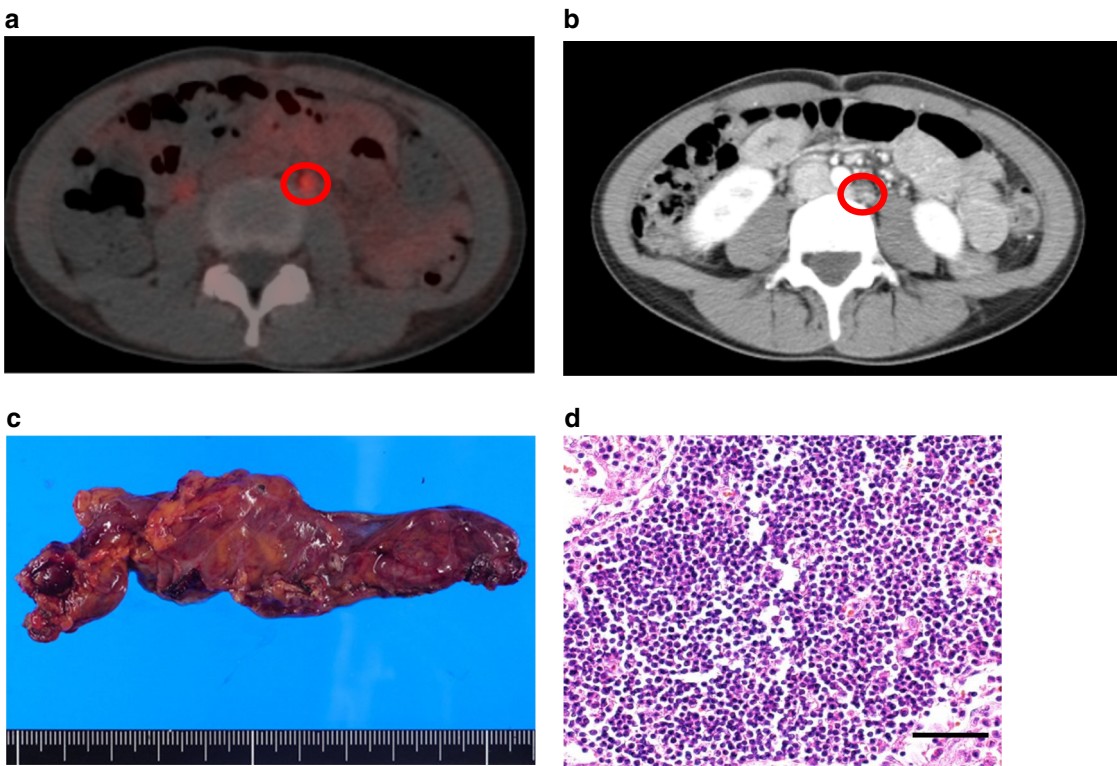

**Fig. 1 A case of cervical cancer in which 18F-FDG-PET/CT produced false-positive results. a, b** Preoperative findings. **a** A transverse 18F-FDG-PET/CT image demonstrating increased 18F-FDG uptake in a left paraaortic lymph node (PALN) (circle). **b** A transverse CT image showing an enlarged left PALN (circle). **c, d** Postoperative findings. **c** Macroscopic appearance of the resected left PALN that had increased 18F-FDG uptake on preoperative 18F-FDG-PET/CT. **d** Representative pathological findings from the resected 38 lymph nodes (hematoxylin and eosin). No tumor cells were noted. Scale bars, 50 mm. The image contains no malignant cells. Scale bar, 50 μm.

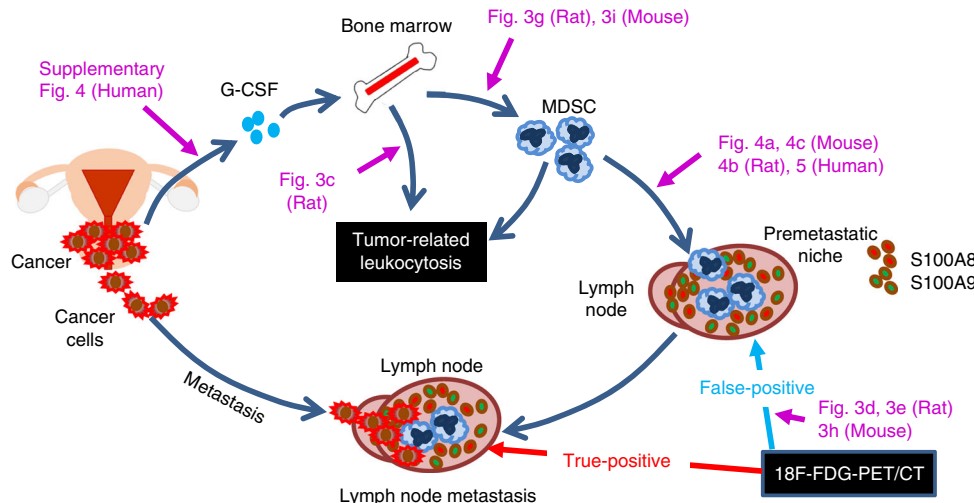

**Fig. 2 Proposed mechanism of false-positive 18F-FDG-PET/CT results in TRL-positive gynecological cancer patients.** MDSC-mediated premetastatic niche formation misleads 18F-FDG-PET/CT during lymph node staging.

validation cohort, 35 patients had cervical, 55 had endometrial, and 35 had ovarian cancer (Supplementary Table 2). As shown in Supplementary Table 3, the sensitivity, specificity, NPV, and PPV of 18F-FDG-PET/CT for lymph node staging were 36.1%, 85.4%, 76.8%, and 50.0%, respectively, which were similar to the results for the primary cohort. Of the patients included in the validation cohort, 17 patients exhibited TRL. As expected, TRL-positive patients showed significantly higher serum G-CSF concentrations than those TRL-negative patients (Supplementary Fig. 4).

When the impact of pretreatment TRL on the diagnostic performance of 18F-FDG-PET/CT in detecting pelvic and paraaortic lymph node metastasis was investigated, as shown in Supplementary Table 4, pretreatment TRL was significantly associated with false-positive 18F-FDG-PET/CT results (TRL vs non-TRL; 35.3% vs 6.5%, $P < 0.0001$).

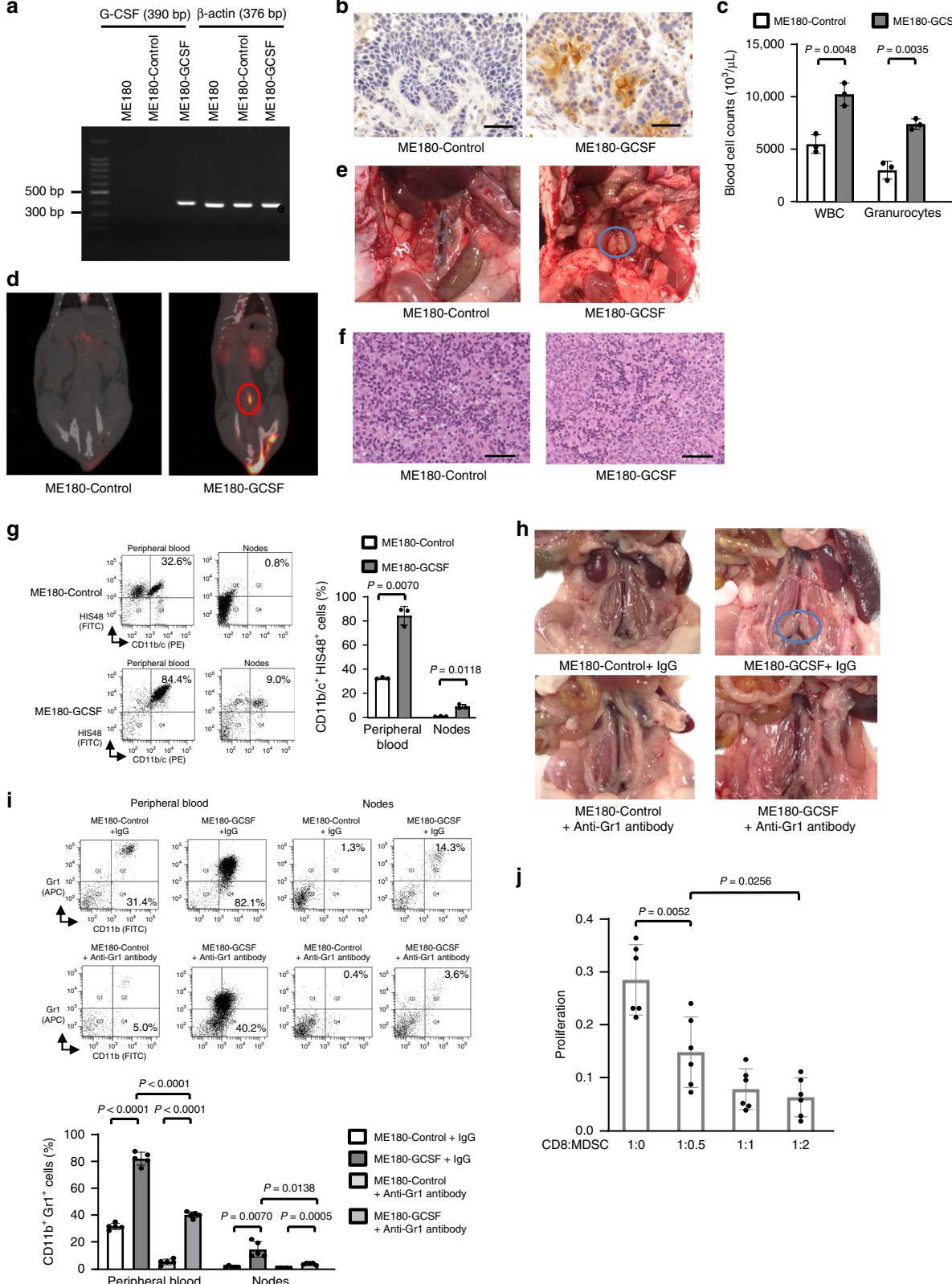

## Discussion

As gynecological cancer frequently disseminates via lymphatic spread, accurate lymph node staging is highly important for determining the optimal treatment strategy.

It has been generally accepted that 18F-FDG-PET/CT has higher diagnostic performance than CT or MRI in detecting metastatic lymph nodes in gynecological cancer patients[26]. Although it has been reported that 18F-FDG-PET/CT reliably detects lymph node deposits with a volume larger than 80 mm³[27], false-positive detection of lymph node involvement has been caused by benign conditions, including systemic inflammation, infectious diseases, or immunizations[28]. In our patients, false-

**Fig. 3 Effects of tumor-derived G-CSF on the PALN and 18F-FDG-PET/CT in animal models of cervical cancer. a** Representative image of G-CSF mRNA expression in ME180-Control or ME180-GCSF cells as evaluated by RT-PCR. **b** Representative G-CSF staining in ME180-Control cells or ME180-GCSF cells-derived tumors. **c** WBC/granulocyte counts of ME180-Control-derived tumor-bearing rats and ME180-GCSF-derived tumor-bearing rats (three rats per group). Rats were subcutaneously inoculated with ME180-Control or ME180-GCSF cells. Three weeks after inoculation, their subcutaneous tumors or peripheral blood samples were collected for analyses. **d** 18F-FDG-PET/CT scan of ME180-Control-derived tumor- and ME180-GCSF-derived tumor-bearing rats. Rats were subcutaneously inoculated with ME180-Control or ME180-GCSF cells. Three weeks after inoculation, 18F-FDG-PET/CT was performed. ME180-GCSF-derived tumor-bearing rats showed significant 18F-FDG-uptake in the PALN (circle). **e** PALNs resected after 18F-FDG-PET/CT. **f** Representative pathological findings from the PALNs resected after 18F-FDG-PET/CT (hematoxylin and eosin). Both of the images contain no malignant cells. **g** Effects of tumor-derived G-CSF on the induction of MDSC in rat models of cervical cancer. CD11b/c$^+$HIS48$^+$ cell populations detected in the peripheral blood and lymph nodes. Rats were subcutaneously inoculated with ME180-Control or ME180-GCSF cells. Three weeks after inoculation, the number of MDSC was evaluated by flow cytometry (three rats per group). **h** Effects of tumor-derived G-CSF and anti-Gr-1 antibody on PALN in mice models of cervical cancer. **i** Effects of anti-Gr-1 antibody on MDSC induction in mice models of cervical cancer. CD11b$^+$Gr1$^+$ cell populations detected in the peripheral blood and lymph nodes. **h, i** ME180-Control- or ME180-GCSF-derived tumor-bearing mice were treated with either control IgG or anti-Gr-1 antibody for three weeks (five mice per group). Then, the number of MDSC was evaluated by flow cytometry (five mice per group). **j** Ability of CD11b$^+$Gr-1$^+$ cells to suppress CD8$^+$ T cell assessed by T-cell proliferation assay. CD11b$^+$ Gr-1$^+$ cells (MDSC) were isolated from spleen of ME180-GCSF-derived tumor-bearing mouse. CD8$^+$ T cells ($2 \times 10^5$ cells/well) were isolated from spleen of Balb/c mice and co-cultured with MDSC at indicated ratios. Cells were incubated for 72 h, after which BrdU was added for an additional 24 h. T cell proliferation was determined by BrdU incorporation ($n = 6$). Error bars indicate mean ± SD. Statistical significance was assessed using two-sided Welch $t$ test. bp, base pairs. Scale bar, 50 μm. Source data are provided as a Source Data file.

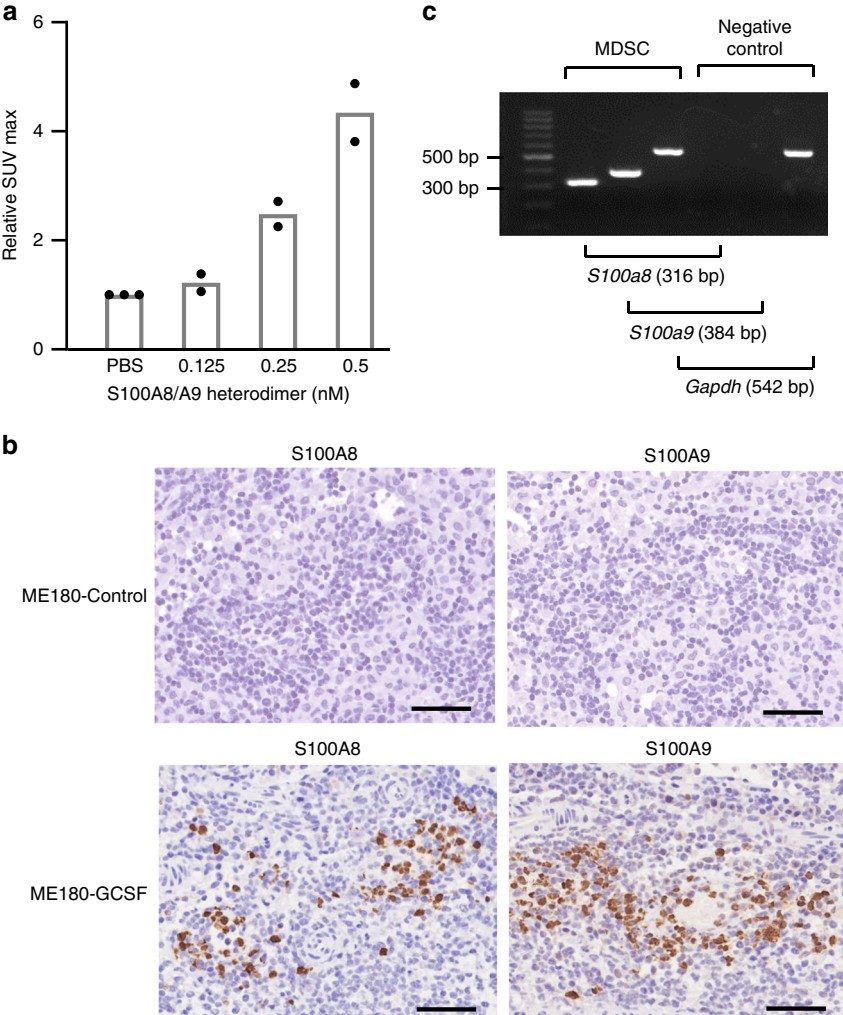

**Fig. 4 S100A8/A9 produced by MDSC induce 18F-FDG-uptake. a** The SUV max of on the site of S100A8/A9 heterodimer injection by 18F-FDG-PET/CT. Mice were subcutaneously injected with indicated concentrations of S100A8/A9 heterodimer. 18F-FDG-PET/CT was performed 24 h after the injection, and the SUV max of the injection site was analyzed (PBS, $n = 3$; others, $n = 2$). **b** Representative immunoreactivities of resected paraaortic lymph nodes for S100A8 and S100A9 (ME180-Control-derived tumor- or ME180-GCSF-derived tumor-bearing rats). Scale bars, 50 μm. **c** Representative image of *S100a8* and *S100a9* mRNA expression in MDSC of ME180-GCSF-derived tumor-bearing mice as evaluated by RT-PCR. bp, base pairs. Source data are provided as a Source Data file.

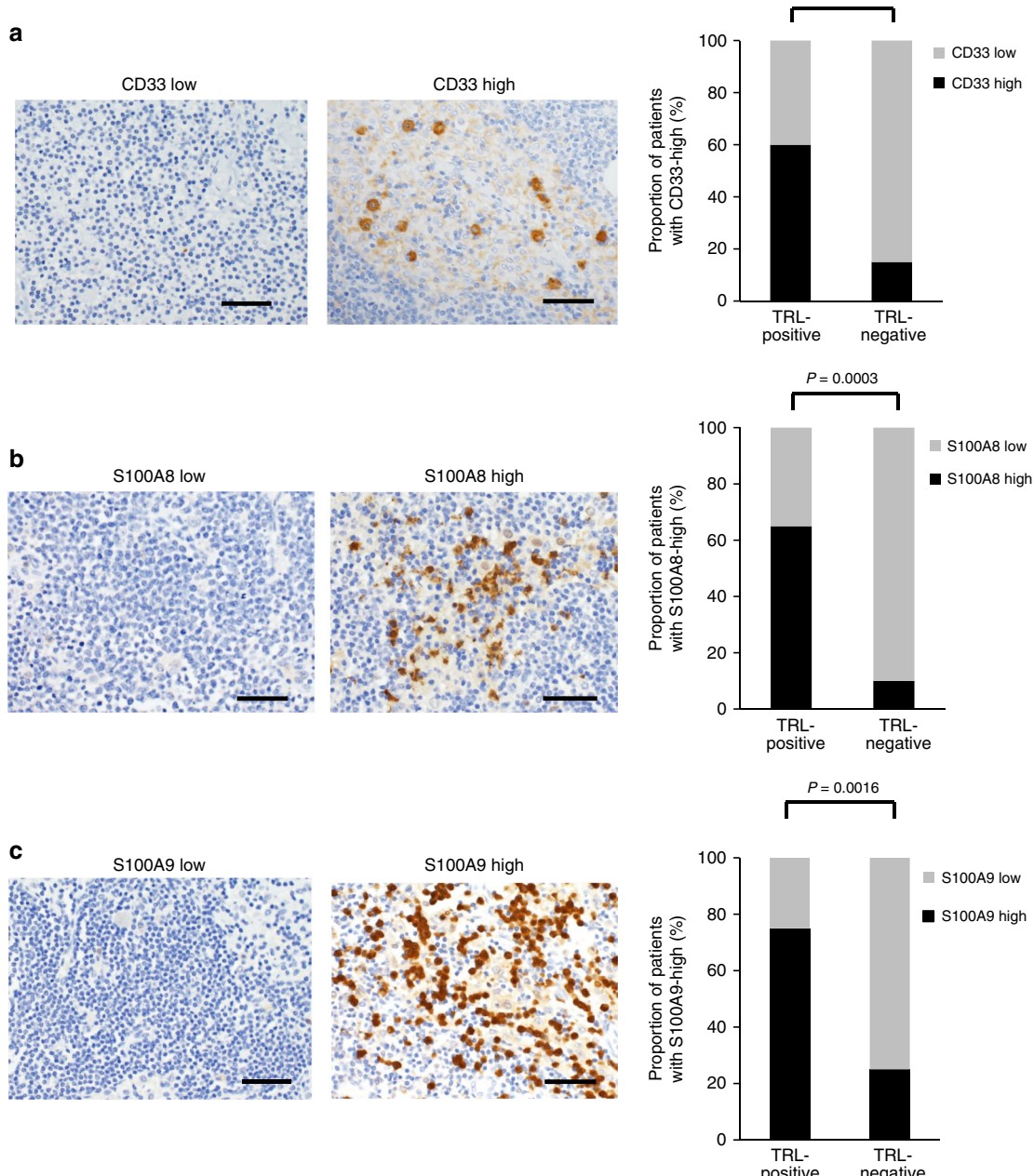

**Fig. 5 Immunoreactivities of resected lymph nodes for CD33 and S100A8/9 in cervical cancer patients TRL-positive versus TRL-negative patients.** **a** CD33 expression in cervical cancer patients according to white blood cell counts. Photographs; representative CD33-stained lymph node sections from cervical cancer patients. A graph; proportion of patients with CD33-high. **b** S100A8 expression in cervical cancer patients according to white blood cell counts. Photographs; representative S100A8-stained lymph node sections from cervical cancer patients. A graph; proportion of patients with S100A8-high. (**c**) S100A9 expression in cervical cancer patients according to white blood cell counts. Photographs; representative S100A9-stained lymph node sections from cervical cancer patients. A graph; proportion of patients with S100A9-high. Scale bars, 50 μm. Statistical significance was assessed using two-sided Fisher's exact test (TRL-positive: WBC ≥ 9000/mL, $n = 20$; TRL-negative: WBC < 9000/mL, $n = 20$). Source data are provided as a Source Data file.

positive lymph node metastasis was observed in 7.9, 5.9, and 8.3% of cervical, endometrial, and ovarian cancer patients, respectively (primary cohort). These false-positive rates are consistent with previously reported data[2–4]. However, most of these false-positive results were not associated with known factors that deteriorate the diagnostic performance of 18F-FDG-PET/CT. Thus, identification of patients or clinical situations where 18F-FDG-PET/CT has a limited diagnostic impact is of great importance.

In the current study, by carefully reviewing the clinical data, we demonstrated that false-positive results occur more frequently in TRL-positive gynecological cancer patients than in TRL-negative patients (Table 2). Importantly, this conclusion was confirmed in the primary cohort, and validated in the validation cohort as well as in animal models of TRL-positive gynecological cancer. This suggests that 18F-FDG-PET/CT has only limited diagnostic performance for lymph node staging of TRL-positive gynecological cancer patients.

Based on our mechanistic investigations, increased false-positive detection of lymph node involvement using 18F-FDG-PET/CT in TRL-positive patients may be explained by MDSC induced by tumor-derived G-CSF and MDSC-mediated pre-metastatic niche formation (Fig. 2). Recently, it has been

demonstrated in the study of antibody-based single-photon emission computed tomography (SPECT), that S100A8/A9 can be a potent imaging biomarker for the detection of MDSC-mediated proinflammatory and immunosuppressive environments in premetastatic tissue (namely, the premetastatic niche)[29]. Moreover, we recently reported that the MDSC-mediated premetastatic niche is responsible for the highly metastatic nature of TRL-positive uterine cervical and endometrial cancer[20]. In the current study, we found that false-positive LNs contained a significant number of MDSC (Fig. 3g) and expressed S100A8/A9 (Fig. 4b), and that tissue S100A8/A9 expression in the absence of cancer cells can induce significant 18F-FDG uptake in the injected tissue (Fig. 4a). Recently, we demonstrated that the MDSC in premetastatic organ secreted the proinflammatory chemokine, chemokine (C-X-C motif) ligand 2 (CXCL2) to attract C-X-C motif chemokine receptor 2 (CXCR2)-expressing cancer cells[20,25]. Collectively, these results strongly indicate that MDSC-mediated premetastatic niche formation in lymph nodes lead to the false-positive 18F-FDG-PET/CT results (Fig. 2).

As shown in Supplementary Fig. 5, among the TRL-positive cervical cancer patients, although false-positive lymph nodes were associated with longer survival when compared with true positive lymph nodes (PFS, $p = 0.0050$; OS, $p = 0.0124$), there were no significant differences between true negative lymph nodes and false-positive lymph nodes (PFS, $p = 0.1757$; OS, $p = 0.3722$). In TRL-positive endometrial and ovarian cancer, no significant differences in survival were observed between patients with true negative nodes, patients with false-positive nodes and those with true negative nodes. These results may indicate that premetastatic niche alone had no negative impact on TRL-positive patient's survival, and that survival rates will be significantly decreased once lymph node metastasis is developed. To avoid the possibility that false-positive detection of distant nodes results in the denial of a chance for surgical or radiological cure in TRL-positive cancer patients, a novel imaging modality that can differentiate the premetastatic niche from healthy or metastatic tissue needs to be developed.

The existence of the premetastatic niche was first demonstrated in 2005 by IHC analyses using patient-derived tumor samples, and its significance in cancer metastasis has been actively investigated[30]. However, due to the difficulty of obtaining premetastatic tissues from patients, most studies conducted have used orthotopic or transgenic mouse cancer models. Although S100A8/A9-specific SPECT imaging succeeded in the detection of premetastatic niche formation in the lungs of tumor-bearing mice[29], the "premetastatic niche" has never been visualized by imaging techniques in pretreatment cancer patients. Thus, to our knowledge, this is the first report demonstrated that the premetastatic niche can be visualized by 18F-FDG-PET/CT in cancer patients.

Our study may have important clinical implications. First, the current study describes a previously undocumented pitfall of 18F-FDG-PET/CT, pretreatment TRL, during lymph node staging in gynecological cancer patients. In the current study, 35.3–41.7% of false-positive 18F-FDG-PET/CT results were associated with TRL (Table 2). Thus, by conducting low-cost peripheral blood examinations, we may be able to identify a group of patients for whom 18F-FDG-PET/CT has limited diagnostic performance during lymph node stating. As TRL is also found in non-gynecological cancer patients, we believe that our findings may significantly aid in the lymph node staging of many human malignancies.

The current study brings further research opportunities. Based on our previous report showing that cervical cancer patients who display tumor-related leukocytosis show increased bone marrow FDG-uptake[31], we investigated whether patients with false-positive LN also had increased bone marrow FDG-uptake. As

shown in Supplementary Fig. 6, in both primary-cohort and validation-cohort, false-positive LN was significantly associated with increased bone marrow FDG-uptake (primary-cohort, $p = 0.0110$; validation-cohort, $p = 0.0049$). These results indicate that, by examining bone marrow FDG-uptake, we may be able to identify patients for whom 18F-FDG-PET/CT is likely to produce false-positive results during lymph node stating. Further researches are required to examine the association between TRL, bone marrow FDG-uptake and false-positive 18F-FDG-PET/CT result in gynecological cancer patients.

The limitations of our study need to be addressed. The first is that our clinical study was conducted at a single institution including a relatively small number of patients. To validate our clinical findings, a collaborative multi-institutional investigation needs to be conducted, preferably in a prospective setting. The second is that we used nude mice/rat inoculated with human cancer cells, as the inoculation of human gynecological cancer cells into immunocompetent mice did not result in tumor development. Third, although CD33 was employed for the identification of human MDSC in the current study based on the previous studies[32,33], CD33-positive cells are not always MDSCs. Thus, to draw a definitive conclusion on the association between MDSC and S100A8/A9, further immunophenotypic characterization of human MDSC in the false-positively detected lymph nodes is required. The fourth is that although the current study focused on the "tumor-derived G-CSF-MDSC axis", other tumor-derived factors may also play roles in the induction of TRL or MDSC in a G-CSF-independent manner. Moreover, we cannot exclude the possibility that other stromal cells in the tumor microenvironment are stimulated by tumor-derived G-CSF to create the premetastatic niche. Accordingly, the mechanism responsible for the false-positive 18F-FDG-PET/CT results should be investigated further. The fifth is our analytical method employed in our clinical study. To evaluate the diagnostic performance of 18F-FDG-PET/CT during lymph node staging, we mainly compared the false-positive results between TRL-positive and TRL-negative patients. However, other methods such as evaluating the diagnostic performance of FDG-PET/CT in TRL-positive and TRL-negative patients using ROC-based methods, might also be useful. Thus, to validate our clinical findings, further investigations preferably in a prospective and multi-institutional setting are required.

In conclusion, we demonstrated that pretreatment TRL is associated with a significantly increased risk of false-positive 18F-FDG-PET/CT results during lymph node staging in gynecological cancer patients. Our mechanistic investigations suggest that MDSC-mediated premetastatic niche formation in response to tumor-derived G-CSF is responsible for the false-positive result. These data warrant further investigation, particularly in multi-institutional and prospective settings.

## Methods

**Patients: primary cohort**. Permission to proceed with data acquisition and analysis was obtained from the institutional review board of Osaka University Hospital. A list of patients with primary, previously untreated, histologically confirmed cervical, endometrial, or ovarian cancer who were treated at Osaka University Hospital between April 2007 and December 2014 was generated from our institutional registry (primary cohort).

Then, through a chart review, 426 patients who had undergone a preoperative 18F-FDG-PET/CT scan before initial staging surgery were identified. Patients who received neoadjuvant chemotherapy or preoperative radiotherapy before 18F-FDG-PET/CT and patients who underwent initial staging surgery without lymphadenectomy were excluded. Patients with coexisting hematologic malignancies, administration of corticosteroids or recombinant G-CSF, or acute or chronic infection were excluded. Patients with sarcoma or carcinosarcoma histology were also excluded. Informed consent was received from all patients and their clinical data were retrospectively reviewed.

**Patients: validation cohort**. The validation cohort included 125 patients with newly-diagnosed gynecological cancers who had undergone a preoperative 18F-FDG-PET/CT scan before initial staging surgery at our institution (from January 2015 to October 2018). Informed consent was received from all patients, and their clinical data were retrospectively investigated.

**Clinical information**. Clinical information regarding demographic or pathological data, oncological and surgical outcomes, and imaging results were collected from medical records. Then, each patient's 18F-FDG-PET/CT, CT of the abdomen and chest, and pelvic MRI results were retrospectively correlated with the histopathological findings of the surgical specimens.

**18F-FDG-PET/CT Protocol**. Informed consent was received from each patient for 18F-FDG-PET/CT scanning. Whole-body imaging using 18F-FDG was carried out with a combined PET/CT scanner (Gemini GXL 16; Philips, Amsterdam, The Netherlands, or SET-3000 GCT/X; Shimadzu, Kyoto, Japan), which provides separate CT and PET datasets that can be accurately fused on a workstation (Voxbase; J-MAC System, Inc, Japan). All patients fasted for at least 4 h prior to the intravenous administration of 18F-FDG at the dose of 3.7 MBq/kg. Whole-body images, generally from the top of the skull to mid-thigh, were acquired in the supine position approximately 60 min after 18F-FDG injection. Image acquisition was initiated with a CT scan for attenuation correction and anatomical localization. The CT scanning parameters were as follows: 120 kV; 60–80 mA; 16 slices; 1.5-mm detector collimation; and 5.0-mm slice thickness. Coronal and sagittal CT images were reconstructed using axial CT images. A whole-body emission PET scan was performed immediately after the CT scan in a three-dimensional mode with a 3.0 min per bed position (11 positions) using a dedicated scanner with 32 rings of bismuth germanate detectors that simultaneously produced 63 slices of 3.125-mm thickness along a 20-cm longitudinal field. Attenuation-corrected PET images were reconstructed by the row-action maximum likelihood algorithm method (RAMLA) in Gemini GXL 16 and dynamic row-action maximum likelihood algorithm (DRAMA) method in SET-3000 GCT/X, respectively.

**Image analysis**. The CT and PET images were transferred to a commercially available workstation (Advantage Windows Workstation, version 4.5; GE Healthcare). All PET/CT images were analyzed by two experienced nuclear medicine physicians. For semiquantitative analysis of 18F-FDG uptake, a volume of interest (VOI) was defined on the target lesions (primary lesion, pelvic or paraaortic lymph nodes, and disseminated sites) on the transaxial PET images. The maximum standardized uptake value (SUVmax) inside a VOI was calculated as follows: concentration of radioactivity in the VOI (MBq/ml) × total body weight (kg)/injected radioactivity (g/MBq). Lymph nodes were considered positive for malignancy when the SUVmax was greater than 2.5, regardless of node size.

**Evaluation of the diagnostic performance of FDG-PET/CT**. Lymph nodes were labeled in seven nodal regions: right and left pelvic, right and left common iliac, presacral, and right and left para-aortic lymph nodes. Right para-aortic lymph nodes included aorto-caval lymph nodes. FDG-PET/CT findings on lymph node metastasis were analyzed on the basis of the pathological findings at the region level, and defined as true-positive, true-negative, false-positive or false-negative. When a patient had false-positive LN and true positive LN in a different region, the FDG-PET/CT result on lymph node metastasis defied as false positive. In like manner, when a patient had false-negative LN and true positive LN in a different region, the FDG-PET/CT result was defied as false negative. Subsequently, sensitivity, specificity, PPV, and NPV for predicting lymph node metastasis were determined.

**Surgical staging and pathological evaluation**. The standard procedures for surgical staging vary by the type of cancer, but are based on the standard treatment guidelines[34–36]. Surgical staging was performed a median of 31 and 30 days after 18F-FDG-PET/CT in the primary cohort and validation cohort, respectively. Pelvic lymphadenectomy aimed to remove external iliac, internal iliac, common iliac, obturator, suprainguinal, and presacral lymph nodes. The para-aortic lymphadenectomy procedure includes the removal of lymph nodes from the mid common iliac artery to at least the inferior mesenteric artery. The resected primary tumor and lymph nodes were sliced and stained with hematoxylin and eosin, and examined microscopically by at least two experienced pathologists. The post-operative histological diagnoses were made according to the criteria of the World Health Organization classification. Patients were staged according to the International Federation of Gynecology and Obstetrics (FIGO) staging criteria.

**Definitions of tumor-related leukocytosis (TRL)**. During the period between the initial presentation of the disease and the day of 18F-FDG-PET/CT, all patients underwent at least one blood test. The lowest leukocyte counts obtained during these tests were used in the current analyses. Pretreatment leukocytosis was defined as the detection of a leukocyte count of ≥9000/µL.

**Cell culture**. ME180 human cervical cancer cell line and Ishikawa human endometrial cancer cell line were purchased from the American Type Culture Collection. Cells were expanded in a humidified at 37 °C with 5% CO₂ in DMEM supplemented with 10% fetal bovine serum (FBS). They were passaged after being received and subsequently stored in liquid nitrogen. Cells were regularly authenticated by examination of morphology and growth characteristics. The frozen stocks were thawed for experiments and used in less than 3 months. They were routinely tested for mycoplasma species (EZ-PCR Mycoplasma Test Kit; Biological Industries; #20-700-20).

**Clone selection**. The expression vector for the mouse G-CSF gene (pCAmG-CSF) and the empty vector (pCAZ 2) used in this study, were provided by the RIKEN BRC through the National Bio-Resource Project run by MEXT. The expression of these genes was driven by the CAG promoter[37,38]. Transfection was performed using Lipofectamine 2000 (Invitrogen; #11668019) according to the manufacturer's instructions. Clonal selection was carried out by adding G418 (Life Technologies; #11811031) to the medium at a final concentration of 500 mg/mL. ME180 cell lines transfected with a G-CSF-expressing vector (ME180-GCSF) or a control vector (ME180-Control) were established. Ishikawa cell lines transfected with a G-CSF-expressing vector (Ishikawa-GCSF) or a control vector (Ishikawa-Control) were also established.

**Immunohistochemical analyses**. To gain further insight into the mechanism of false-positive 18F-FDG-PET/CT results, immunohistochemical analyses were performed. Surgical specimens were fixed in 10% neutral buffered formalin, embedded in paraffin, sectioned, and processed for IHC staining. The primary antibodies used were anti-G-CSF polyclonal antibody (Santa Cruz Biotechnology; #SC-1318), anti-CD33 antibody (Leica Biosystems Inc; #NCL-L-CD33), anti-S100A8 antibody (human, Abcam; #ab92331, rat, BOSTER BIOLOGY; #PB9742), and anti-S100A9 antibody (NOVUS Biologicals; #NB110-89726). The surrounding nonneoplastic stroma served as an internal negative control for each slide. Optical images were captured using PROVIS AX80 (Olympus).

**Flow cytometry**. Single-cell suspensions were prepared from rat blood and lymph nodes. RBCs were removed using ammonium chloride lysis buffer. Cells were filtered through 40-µm nylon strainers, incubated with antibodies, and analyzed by flow cytometry. Based on previous reports that rat MDSC express CD11b/c and HIS48[18,19], the following labeled monoclonal antibodies were used for the staining of rat MDSC: anti-rat PE-conjugated anti-CD11b/c (Miltenyi Biotec; #REA325) and anti-rat FITC-conjugated anti-HIS48 (Invitrogen; #11-0570-82). The following labeled monoclonal antibodies were used for the staining of moue MDSC: anti-human/mouse FITC-conjugated anti-CD11b (Tonbo Biosciences; #35-0112) and anti-mouse APC-conjugated anti-Ly6G (Gr1) (Tonbo Biosciences; #5931). Flow cytometric data were acquired on FACSCanto II flow cytometer and analyzed using FACSDiva software (BD Biosciences, San Jose, CA). Cells that had been incubated with irrelevant isotype-matched antibodies and unstained cells served as controls.

**Animal experiments**. All procedures involving mice and rats, and their care were approved by the Institutional Animal Care and Use Committee of the Osaka University in accordance with the relevant institutional and NIH guidelines (Approved No; 29-021-002).

Initial experiments were conducted to examine whether false-positive 18F-FDG-PET/CT results were frequently observed in the animal model of TRL-positive gynecological cancer. Briefly, 7- to 8-week-old female F344/NJcl-rnu/rnu rats (NIH-RNU; Japan SLC, Shizuoka, Japan) were subcutaneously inoculated with either $2 \times 10^7$ ME180-Control or ME180-GCSF cells in 200 µl of PBS into their right flanks. Three weeks after the inoculation, the rats were anesthetized by intraperitoneal injection of a mixture of medetomidine, midazolam, and butorphanol, or inhalation of 2% isoflurane and 18F-FDG was injected into the tail vein. Sixty minutes after the injection, the PET measurement was performed in an abdominal position using a small-animal PET-CT scanner (Inveon MM; Siemens Medical Solutions, Knoxville, USA). The CT scan was performed before the PET scan to determine the exact position of paraaortic lymph nodes. The image data acquired from the small-animal PET-CT scanner were displayed and analyzed with RadiAnt DICOM 4.6.9 Viewer (Medixant, Poznan, Poland). Rats were sacrificed after completion of scanning, para-aortic lymph nodes were resected, fixed in formalin, paraffin-embedded, and subjected to histological or immunohistochemical analyses.

The second experiment was conducted to examine the effects of subcutaneously inoculated G-CSF expressing cervical cancer on the size of paraaortic lymph nodes. Briefly, 5- to 6-week-old BALB/c nude mice were subcutaneously inoculated with either $1 \times 10^7$ ME180-Control or ME180-GCSF cells in 150 µL of PBS into their right flanks. Control IgG or anti-Gr-1 antibody (BioXCell, #BE0075) was intraperitoneally administered to ME180-GCSF-bearing mice at a dose of 200 µg/mouse every 48 h starting 1 day after inoculation. Three weeks after the inoculation, the mice were sacrificed for the evaluation of para-aortic lymph node metastasis.

**Isolation of MDSC**. MDSC were isolated from single-cell preparations of mouse splenocytes using a Myeloid-Derived Suppressor Cell Isolation Kit (mouse) and an MS column (Miltenyi Biotec; #130-042-201), according to the manufacturer's instructions. The purity of the isolated cell populations was determined by flow cytometry, and the frequency of CD11b⁺Gr-1⁺ cells was >95%[31].

**T cell proliferation assay**. A 96-well plate was coated with 1 μg/well of anti-CD3e antibody (Tonbo Biosciences, #70-0031). CD8 positive T cells were purified from the spleen of Balb/c mice using T cell isolation columns (R&D Systems; #MAGM203) according to the manufacturer's instructions. To determine the impact of MDSC on T cell proliferation, purified MDSC from the spleen of ME180 GCSF-derived tumor-bearing mouse were co-cultured with T cells. Cell pro-liferation was assayed using a cell proliferation ELISA BrdU kit (Roche Applied Science; #11647229001).

**Reverse transcriptase PCR**. RNA was extracted from MDSC using TRIzol (Life Technologies; #15596018). The resultant total RNA (1 μg) was used to synthesize cDNA with ReverTraAce qPCR RT Master Mix (Toyobo; #FSQ-201). PCR was performed using TaqMan PCR master mix (Qiagen; #201443) and specific primers. The PCR primers were purchased from Life Technologies. The sequences of the primers used were as follows: human β-actin: forward primer, 5′-CGTGACATT AAGGAGAAGCTGTG-3′ and reverse primer, 5′-GCTCAGGAGGAGCAATGAT CTTGA-3′; human G-CSF: forward primer, 5′-TGAGTGTGCCACCTACAAGC-3′ and reverse primer, 5′-GACACCTCCAGGAAGCTCTG-3′, mouse glyceraldehyde 3-phosphate dehydrogenase (*Gapdh*): forward primer, 5′-TTAGCCCCCCTGGCC AAGG-3′ and reverse primer, 5′-CTTACTCCTTGGAGGCCATG-3′; mouse *S100a8*: forward primer, 5′-TCGTGACAATGCCGTCTGAA-3′ and reverse primer, 5′-GACATATCCAGGGACCCAGC-3′; mouse *S100a9*: forward primer, 5′-AGATG GCCAACAAAGCACCT-3′ and reverse primer, 5′-TAGACTTGGTTGGGCAGC TG-3′. Amplification was performed using a Takara PCR personal-type thermal cycler (Takara).

**Enzyme-linked immunosorbent assay (ELISA)**. The concentrations of human G-CSF were measured using a Human G-CSF Quantikine ELISA Kit (R&D Systems; #DCS50). Absorbance values were measured using a microplate reader (iMark Microplate Reader; Bio-Rad Laboratories, Inc., Hercules, CA).

**Statistics and reproducibility**. Data are representative of at least three independent experiments. Results are expressed as the mean ± SD. Continuous data were compared between the groups using the two-sided Welch $t$ test or two-sided Wilcoxon rank-sum test. Frequency counts and proportions were compared between groups using the two-sided Fisher's exact test. Multiple logistic regression analysis was performed to investigate the factors leading to the false-positive 18F-FDG-PET/CT results. $P$-values of <0.05 were considered significant. All analyses were performed using JMP® software, version 14.0 (SAS Institute. Inc, Cary, NC, USA) or GraphPad Prism version 8.00 (GraphPad Software, La Jolla, California, USA).

**Reporting summary**. Further information on research design is available in the Nature Research Reporting Summary linked to this article.

## Data availability

The source data underlying Figs. 3a, 3c, 3g, 3i, 3j, 4a, 4b and 5, and Supplementary Figs. 1a, 1c, 1g, 2, 3 and 4 are provided as a Source Data file. All other data that support the findings of this study are available within the article and its Supplementary Information files or from the corresponding author upon reasonable request.

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

## Acknowledgements
The authors thank Hirofumi Hamada (Sapporo Medical University) for providing them with information regarding the G-CSF expression plasmids. The authors' research was supported by grants-in-aid for General Scientific Research T17K16849, A15H025640, and T17K112760 from the Ministry of Education, Culture, Sports, Science, and Technology of Japan.

## Author contributions
S.M. designed the experiments, acquired the clinical data and wrote the manuscript; N.K. designed and performed the experiments, contributed data analysis and wrote the paper; T.S. designed the experiments and edited the paper; K.S., E.Y., and K.K. performed the experiments and acquired the clinical data; H.K., R.T., M.K., and Y.M. designed the experiments; H.K., J.H., and T.K. edited the paper.

## Competing interests
The authors declare no competing interests.
