## [Peer Review File · Nature Communications]

Reviewers' comments:

Reviewer #1 (Remarks to the Author): Expert in leukocytosis

This work is performed by authors who previously demonstrated the adverse role of GM-CSF tumor production, stimulating the myeloid compartment eventually leading to a negative, tumor-promoting role of neutrophilia.

The work is based on a significant series of patients for which there is a clear clinical, imaging and pathological correlation underscoring the limits of PET/CT for staging of GY malignancies.

They showed that false positive lymph node visualized on PET correlate with a proliferative activity within the lymph node which corresponds to pre-metastatic niche formation.

These data are of clear importance given the direct implication for tumor staging (and not "stating" as written into the title), using multivariate analysis, the authors show that hyperleukocytosis correlates with the likelihood of false positivity.

The murine findings, GM-CSF engineered tumors correlate with lymph node enlargement and signs of pre-metastatic niche formation parallels the observation in patients. The methodology used is robust and straightforward, and the claims are supported by data.

One aspect that could increase the relevance and impact of the manuscript is tumor biology, are the tumors with false positive lymph node and hyperleukocytosis all related to GM-CSF production, and did these patients present a particularly worse outcome? If so, is the outcome driven solely by hyperleukocytosis?

Another comment of importance, it was recently shown that bone marrow FDG fixation correlates with outcome and hyperleukocytosis, it would be of interest to have this information in the current series.

Reviewer #2 (Remarks to the Author): Expert in PET imaging in gynecological cancers

Comments to false positive PET in CC

This manuscript presented a very interesting association of pretreatment tumor-related leukocytosis (TRL) with false-positive FDG-PET/CT results in lymph node staging/detection in gynecological cancer patients. The study included a retrospective analysis of 426 patients (between April 2007 and December 2014, including 127 cervical cancer, 203 endometrial cancer, and 96 ovarian cancer patients) who had undergone a preoperative FDG-PET/CT and a staging surgery (performed at a median of 30 days after FDG-PET/CT), a prospective validation cohort of 125 patients (from January 2015 to October 2018, including 35 cervical cancer, 55 endometrial cancer, and 35 ovarian cancer patients), and rat and mouse models of TRL-positive and TRL-negative cervical cancer (Figures 2 and 3), and immunohistochemical analyses (CD33 and S100A8/9) of dissected lymph nodes from cervical cancer patients.

For this manuscript, I have some suggestions/comments for further revisions:

1. The results of preoperative FDG-PET/CT in the manuscript seemed to be of patient-based interpretation. However, the number of false-positive or false-negative lymph nodes can be multiple and they may be present in the same patient. A more elaborate analysis using lesion-based or nodal region-based interpretation should be performed.
2. The authors have investigated the diagnostic performance of FDG-PET/CT in detecting pelvic and paraaortic lymph node metastasis. The diagnostic performance should be also presented and compared using the ROC-based methods, in addition to the comparison of false-positive results.
3. To validate the association of pretreatment TRL with false-positive lymph node staging by FDG-PET/CT, the authors prospectively enrolled newly-diagnosed gynecological cancer patients with informed consents. For a prospective study, the detailed inclusion and exclusion criteria, the study

flow diagram (e.g. CONSORT diagram), and the IRB approval document should be provided. In addition, have you measured serum G-CSF levels in this prospective study? What is the difference of serum G-CSF levels between patients with and without TRL?

4. The authors concluded that the MDSC-mediated premetastatic niche created in the lymph node of TRL-positive patients misleads FDG-PET/CT for detecting nodal metastasis. To support this conclusion, immunohistochemical analyses have been performed in the dissected lymph nodes obtained from patients. To establish more direct evidence for the conclusion, the analytical results of false-positive lymph nodes detected by FDG-PET/CT should be presented. How many false-positive lymph nodes in patients were dissected and collected? How many of these lymph nodes have undergone related immunohistochemical analyses? What are the results?

Reviewer #3 (Remarks to the Author): Expert in MDSC

The manuscript describes a novel phenomenon of false-positive diagnostic imaging of cervical cancer due to the uptake of 18F-FDG by tumor infiltrating MDSC. Using a single rodent cervical cancer model, ME-180 and the engineered, G-CSF expressing counterpart cell line, authors have demonstrated 18F-FDG signal was detected in paraaortic lymph node (PALN) which contains increased frequency of S100a8/9-producing MDSC. Infiltration of MDSCs in the PALNs is driven by the pro-inflammatory cytokines such as G-CSF. Based on these data they conclude that one explanation of the misleading diagnosis of metastatic cervical cancer is the 18F-FDG uptake by MDSC during pre-metastatic stage.

Major Comments

1) The entire study is based on a single rodent cervical cancer model. We would like to see if the data can be reproduced in other clinical relevant model systems in order to exclude the findings are specific to ME-180 cell line.

2) The data of Fig 2G is not convincing as the authors have only used ME180-G-CSF tumor bearing mice for injection of anti-Gr1 antibody, whereas the control ME180 tumor bearing mice were not treated in the same way. Moreover, histological staining data should be provided to demonstrate depletion of MDSC in PALNs by anti-Gr1 antibody.

3) Likewise, in Fig 2H to draw the conclusion of infiltration of MDSC into PALNs by G-CSF, authors need to provide MDSC frequency data in PALNs from ME180 control mice. If the nodes are not available for evaluation, author needs to further optimize the model system.

4) Fig 3B provided here is from ME180-G-CSF tumor bearing rats; however the data of S100a8 and S100a9 in ME180-control tumor bearing rat is missing. We would appreciate if the data could be aligned, as the data (Fig 2 and 3) has been interchangeably presented between mouse and rat model. A clear schematic diagram for either of the model system is required without jumping from one model system to other.

5) Functional immune suppressive studies need to be done to validate the identification of MDSC from rat or mouse.

6) The link between CD33 and S100a8/S100a9 can only be established with help of proper human MDSC markers (CD11b+/CD33+/HLADR-/low/CD14+ or CD15+) which are missing in the manuscript. Further, immunophenotypic characterization of human MDSC infiltrating in draining lymph nodes of cervical cancer patients for the uptake of 18F-FDG is required.

We thank the Reviewers for carefully reading our manuscript and providing us with useful comments. Our manuscript was revised based on the reviewers' comments, and a detailed response to each point is presented below.

Responses to Reviewer #1 (Expert in leukocytosis):

Comment 1:

One aspect that could increase the relevance and impact of the manuscript is tumor biology, are the tumors with false positive lymph node and hyperleukocytosis all related to GM-CSF production, and did these patients present a particularly worse outcome? If so, is the outcome driven solely by hyperleukocytosis?

Responses:

Thank you for the reviewer's thoughtful comments. We have investigated the G-CSF immunoreactivity for all primary tumor in patients with false positive lymph node. As shown in Supplemental Figure 3, among the patients with false positive lymph node, TRL-positive patients showed significantly higher G-CSF expression than TRL-negative patients. We have described this in lines 181-183 of the revised manuscript. We also investigated the survival of patients with false positive lymph node and leukocytosis. As shown in Supplemental Figure 5, among the TRL-positive cervical cancer patients, false positive lymph nodes were associated with longer survival, when compared with true positive lymph nodes (PFS, $p=0.0050$; OS, $p=0.0124$). Moreover, there were no significant differences between true negative lymph nodes and false positive lymph nodes (PFS, $p=0.1757$; OS, $p=0.3722$). In TRL-positive endometrial and ovarian cancer, no significant differences in survival were observed between patients with true negative nodes, patients with false positive nodes and those with true negative nodes. These results may indicate that premetastatic niche alone had no negative impact on patient's survival, and that survival rates will be significantly decreased once lymph node metastasis is developed. We have described this in lines 235-245 of the revised manuscript.

Comment 2:

Another comment of importance, it was recently shown that bone marrow FDG fixation correlates with outcome and hyperleukocytosis, it would be of interest to have this information in the current series.

Responses:

Thank you for the reviewer's comment. I totally agree that it is of great interest. We had shown that increased bone marrow FDG uptake is frequently observed in cancer patients who exhibit leukocytosis (J Natl Cancer Inst 2014;106, pii: dju147). Since then, we have been investigating the association between bone marrow FDG-uptake and the prognosis of all gynecological cancer patients.

The project has been completed and we are almost ready for submission. As this is a big and different project, we cannot include the data in the current study. Please understand our situation.

Responses to Reviewer #2 (Expert in PET imaging in gynecological cancers):

Comment 1:

The results of preoperative FDG-PET/CT in the manuscript seemed to be of patient-based interpretation. However, the number of false-positive or false-negative lymph nodes can be multiple and they may be present in the same patient. A more elaborate analysis using lesion-based or nodal region-based interpretation should be performed.

Responses:

Thank you for the reviewer's thoughtful comments. Our original analyses were not patient-based interpretation, but region-based interpretation. To avoid confusion, we have indicated this in lines 342-347 of the revised manuscript.

Comment 2:

The authors have investigated the diagnostic performance of FDG-PET/CT in detecting pelvic and paraaortic lymph node metastasis. The diagnostic performance should be also presented and compared using the ROC-based methods, in addition to the comparison of false-positive results.

Responses:

Thank you for the reviewer's thoughtful comment. To evaluate the diagnostic performance of 18F-FDG-PET/CT during lymph node staging using the ROC-based methods, SUV max for lymph nodes are required. However, in our hospital, radiologists measured SUV max only for the suspected lymph nodes which showed increased FDG-uptake relative to the uptake in comparable normal structures or surrounding tissue. Thus, our PET results show only specific value (e.g. SUV max of 2.9) for suspected nodes or "no significant uptake" for others. So, in the majority of cases, SUV max of the lymph nodes were not available. To perform re-analyses of SUV max for the nodes for research purpose, we need to take appropriate informed consent from patients and approval of IRB, which require very long time and is not realistic. However, as this is important, we have included a discussion regarding this issue in the revised manuscript (lines 276-282).

Comment 3:

To validate the association of pretreatment TRL with false-positive lymph node staging by FDG-PET/CT, the authors prospectively enrolled newly-diagnosed gynecological cancer patients with informed consents. For a prospective study, the detailed inclusion and exclusion criteria, the study flow diagram (e.g. CONSORT diagram), and the IRB approval document should be provided. In addition, have you measured serum G-CSF levels in this prospective study? What is the difference of serum G-CSF levels between patients with and without TRL?

Responses:

I am sorry for the confusion, due to our English problem. Using the clinical data obtained from gynecological cancer patients who were diagnosed between April 2007 and December 2014

(primary-cohort), we had conducted the analysis in January 2015. Then, we felt the need for a validation of the results from primary-cohort using patients with newly-diagnosed gynecological cancer. The patients who were diagnosed with gynecological cancers between January 2015 and October 2018 were included in the validation-cohort, and their clinical data were retrospectively analyzed. Thus, “prospectively enrolled” is not correct. We have changed the wordings (line 302-306 of the revised manuscript).

We tried to measure the pretreatment serum G-CSF levels in the validation-cohort. However, in TRL-positive cases, only 12 blood samples were available. Thus, we have measured 24 samples (12 TRL-positive and 12 TRL-negative cases), and provided the results as Supplemental Figure 4. As shown, the serum G-CSF level was significantly elevated in patients who display TRL than those without TRL (lines 192-194 of the revised manuscript).

Comment 4:

The authors concluded that the MDSC-mediated premetastatic niche created in the lymph node of TRL-positive patients misleads FDG-PET/CT for detecting nodal metastasis. To support this conclusion, immunohistochemical analyses have been performed in the dissected lymph nodes obtained from patients. To establish more direct evidence for the conclusion, the analytical results of false-positive lymph nodes detected by FDG-PET/CT should be presented. How many false-positive lymph nodes in patients were dissected and collected? How many of these lymph nodes have undergone related immunohistochemical analyses? What are the results?

Responses:

Of the 40 lymph nodes immunohistochemically examined in Figure 4A, nine were false-positive lymph nodes. Of these, more than 75% showed strong immunoreactivities for CD33, S100A8, S100A9 (Supplemental Figure 2), which were consistent with the results obtained in animal investigations. We have described these in the revised manuscript (lines 178-181).

Responses to Reviewer #3 (Expert in MDSC):

Comment 1

The entire study is based on a single rodent cervical cancer model. We would like to see if the data can be reproduced in other clinical relevant model systems in order to exclude the findings are specific to ME-180 cell line.

Responses:

To exclude the possibility that findings are specific to ME-180 cell line, we have conducted additional experiments using Ishikawa cell line. The results were shown in the revised Supplemental Figure 1 and explained in lines 168-171 of the revised manuscript.

Comment 2

The data of Fig 2G is not convincing as the authors have only used ME180-G-CSF tumor bearing mice for injection of anti-Gr1 antibody, whereas the control ME180 tumor bearing mice were not treated in the same way. Moreover, histological staining data should be provided to demonstrate depletion of MDSC in PALNs by anti-Gr1 antibody.

Responses:

We have treated ME180-Control tumor bearing mice with anti-Gr1 antibody, and the results were shown in the revised Figure 3H and explained in lines 140-147 of the revised manuscript. Moreover, to demonstrate the depletion of MDSC in PALNs by anti-Gr1 antibody, we tried to obtain the staining results. However, PALNs obtained from the ME180-Control tumor bearing mice were too small to perform immunohistochemical staining. Thus, we performed flow cytometrical analyses, and provided the results in the revised Figure 3I. The results section was revised accordingly (in lines 147-150 of the revised manuscript).

Comment 3:

Likewise, in Fig 2H to the draw the conclusion of infiltration of MSDC into PALNs by G-CSG, authors need to provide MDSC frequency data in PALNs from ME180 control mice. If the nodes are not available for evaluation, author needs to further optimize the model system.

Responses:

By repeatedly conduct mice experiments, we obtained PALNs from ME180-Control derived tumor bearing mice. We examined the MDSC frequency in these PALNs, and the results were shown in the revised Figure 3I (in lines 147-150 of the revised manuscript).

Comment 4:

Fig 3B provided here is from ME180-G-CSF tumor bearing rats; however the data of S100a8 and S100a9 in ME180-control tumor bearing rat is missing. We would appreciate if the data could be aligned, as the data (Fig 2 and 3) has been interchangeably presented between mouse and rat model.

A clear schematic diagram for either of the model system is required without jumping from one model system to other.

Responses:

As suggested, we have included the IHC data (S100a8 and S100a9) in ME180-Control tumor bearing rat in the revised Figure 4B, and explained in lines 160-163 of the revised manuscript.

It was hard to align the Figures. However, to avoid confusion, a schematic diagram showing the proposed mechanism and the corresponding figures has been included in the revised version (Figure 2).

Comment 5:

Functional immune suppressive studies need to be done to validate the identification of MDSC from rat or mouse.

Responses:

To demonstrate the suppressive activity of MDSCs obtained from our experimental models, we have conducted a T cell proliferation assay. The result was shown in the revised Figure 3J, and explained in lines 150-152 of the revised manuscript.

Comment 6:

The link between CD33 and S100a8/S100a9 can only be established with help of proper human MDSC markers (CD11b+/CD33+/HLADR-/low/CD14+ or CD15+) which are missing in the manuscript. Further, immunophenotypic characterization of human MDSC filtering in draining lymph nodes of cervical cancer patients for the uptake of 18F-FDG is required.

Responses:

As the reviewer pointed, although CD33 was employed for the identification of human MDSCs in the current study, CD33-positive cells are not always MDSCs. There are 3 reasons for such experimental design. The first is that that multiple staining such as CD11b+/CD33+/HLADR-/low/CD14+ or CD15+ is impossible by IHC. The second is that CD33 has been employed as a MDSC marker in IHC experiments (Cancer Res 2016;76:3156-65, Immunity 2013;39:611-21). The last is that although we can analyze CD11b+/CD33+/HLADR-/low/CD14+ or CD15+ cells by FACS, fresh LN samples are needed for the analyses (It takes some more years to obtain LNs from patients with or without false positive PET results). As this is important, we have included a short discussion on this issue (in lines 267-271 of the revised manuscript).

Reviewers' comments:

Reviewer #1 (Remarks to the Author):

Since the authors did previously published on the adverse role of leukocytosis and MDSCs in the same model, I believe that the metabolic activity of bone marrow assessed by PET should really be displayed in parallel, in particular it would be of utmost interest to see if intense bone marrow activity, leukocytosis and false positive lymph nodes occur in the same patients, this would be an informative information since the authors have a track record based on the adverse role of MDSCs.

Reviewer #2 (Remarks to the Author):

There is one issue remained. Although the authors said that the results of preoperative FDG-PET/CT were region-based interpretation (classified into seven nodal regions), the results shown in table 2 and supplemental tables 1 and 3 are evidently patient-based interpretation. If the authors classify a patient's mixed FP and FN results in lymph node regions based on only the presence of FP results for table 2 and supplemental tables 1 and 3, they should address such in the methods.

Reviewer #3 (Remarks to the Author):

The authors have conducted additional experiments which satisfactorily address all of my previous concerns.

We thank the Reviewers for carefully reading our manuscript and providing us with useful comments. Our manuscript was revised based on the reviewers' comments, and a detailed response to each point is presented below.

Responses to Reviewer #1

Comments:

Since the authors did previously published on the adverse role of leukocytosis and MDSCs in the same model, I believe that the metabolic activity of bone marrow assessed by PET should really be displayed in parallel, in particular it would be of utmost interest to see if intense bone marrow activity, leukocytosis and false positive lymph nodes occur in the same patients, this would be an informative information since the authors have a track record based on the adverse role of MDSCs.

Responses:

Thank you for your thoughtful comment. We had shown cervical cancer patients who display tumor-related leukocytosis show increased bone marrow FDG-uptake (*J Natl Cancer Inst* **106**, pii: dju147, 2014). Thus, theoretically, as pointed by the reviewer, there should be a significant correlation between false-positive LN and bone marrow FDG-uptake. In the revised manuscript, using clinical data obtained from primary-cohort and validation-cohort, we have investigated whether patients with false-positive LN also had increased bone marrow FDG-uptake. As shown in the revised Supplemental Figure 6, a result, we have found that false-positive LN was significantly associated with increased bone marrow FDG-uptake (primary-cohort, $p=0.0110$; validation-cohort, $p=0.0049$). Manuscript were revised accordingly (lines 262-271 of the revised manuscript). To avoid the duplicated publication, we did not include the data showing the association between bone marrow FDG-uptake and tumor-related leukocytosis in the current study.

Responses to Reviewer #2

Comments:

Although the authors said that the results of preoperative FDG-PET/CT were region-based interpretation (classified into seven nodal regions), the results shown in table 2 and supplemental tables 1 and 3 are evidently patient-based interpretation. If the authors classify a patient's mixed FP and FN results in lymph node regions based on only the presence of FP results for table 2 and supplemental tables 1 and 3, they should address such in the methods.

Responses:

Thank you for the reviewer's thoughtful comment. As described in the Methods, basically, FDG-PET/CT findings on lymph node metastasis were analyzed on the basis of the pathological findings at the region level, and defined as true-positive, true-negative, false-positive or false-negative. However, when a patient had false positive LN and true positive LN in a different region, the FDG-PET/CT result on lymph node metastasis defined as false positive. In like manner, when a patient had false negative LN and true positive LN in a different region, the FDG-PET/CT result was defined as false negative. The number of such patients were 21 (out of 426) in the primary cohort and 7 (out of 125) in the validation cohort. We have described these in lined 356-359 of the revised manuscript.